# Consonant is all you need: a compact representation of English text for efficient NLP

**Maged S. Al-Shaibani** and **Irfan Ahmad**
ICS Department, KFUPM, Dhahran 31261, Saudi Arabia
SDAIA–KFUPM Joint Research Center for AI, Dhahran 31261, Saudi Arabia
g201381710@kfupm.edu.sa, irfan.ahmad@kfupm.edu.sa

## Abstract

In natural language processing (NLP), the representation of text plays a crucial role in various tasks such as language modeling, sentiment analysis, and machine translation. The standard approach is to represent text in the same way as we, as humans, read and write. In this paper, we propose a novel approach to represent text with only consonants which presents a compact representation of English text that offers improved efficiency without sacrificing performance. We exploit the fact that consonants are more discriminative than vowels and by representing text using consonants, we can significantly reduce the overall memory and compute footprint required for storing and processing textual data.

We present two alternative representations: 'consonants-only', where we completely remove the vowels from the text, and 'masked-vowels', where we mask all the vowels into one special symbol. To evaluate our approaches, we conducted experiments on various NLP tasks, including text classification, part-of-speech (POS) tagging, named-entity recognition (NER), and neural machine translation (NMT), in addition to language modeling. Our results demonstrate that the proposed consonant-based representation achieves comparable performance compared to the standard text representation while requiring significantly fewer computational resources. Furthermore, we show that our representation can be seamlessly integrated with existing NLP models and frameworks, providing a practical solution for efficient text processing.

Last but not the least, we present a technique to retrieve the vowel information from our processed text representation keeping in mind the need to reproduce text in human readable form in some NLP applications. The experiments implementation for this study is made publicly available[1].

---

[1] https://github.com/MagedSaeed/EnglishConsonants

## 1 Introduction

In the field of Natural Language Processing (NLP), text representation and tokenization are closely intertwined and have significant implications for various aspects of an NLP task, such as the vocabulary size, embedding size, and out-of-vocabulary (OOV) rates. The standard approach to text representation in English involves including both consonants and vowels, mirroring the way humans read and write. However, this representation may not be an optimal representation for English NLP. In this paper, we propose a novel approach to English text representation in NLP, inspired by the Abjad writing system, where only consonants are considered as characters and vowels can be inferred (cf. (Daniels, 2013)). Our objective is to develop a compact and efficient representation of English text that not only reduces computational demands but also impacts key aspects of NLP tasks. By focusing solely on consonants, we aim to achieve a significant reduction in vocabulary size, resulting in smaller embedding layers and lower OOV rates. We present two alternative representations: the "consonants-only" approach, where vowels are entirely removed from the text, and the "masked-vowels" approach, where all the vowels are masked to a special symbol. By leveraging the inherent discriminative nature of consonants, which carry more linguistic information compared to vowels, we can substantially decrease the memory and compute footprint associated with storing and processing textual data. To evaluate the effectiveness of our proposed consonant-based representations, we conducted experiments on various NLP tasks, including text classification, POS tagging, named-entity recognition (NER), neural machine translation (NMT), and language modeling. Our results demonstrate that our approach not only achieves comparable performance to the standard text representation but also leads to significant reductions

in vocabulary size, resulting in smaller embedding layers and lower OOV rates. These findings highlight the practical benefits of our approach in terms of memory efficiency and improved handling of unseen words. Furthermore, we address the need for human readability in certain NLP applications, such as machine translation and question answering, by introducing a technique to retrieve the vowel information from our consonants-only text representations. This ensures that our approach remains practical and applicable in scenarios where reproducing text in its original form is necessary. The contributions of the paper can be summarized as below:

- We present novel representations of English text for NLP where the vowel characters are either completely removed or masked to one special symbol. This leads to significant reduction in vocabulary size, embedding layers, OOV rates, and training times.

- A statistical n-gram language model, as well as two neural language models, RNN-based and transformer-based, were built and analyzed to study and compare the new representations.

- We performed several NLP tasks such as sentiment analysis, text classification, POS tagging, NER and NMT using the standard text representation and our two proposed representations and compared their performances.

- We present and evaluate a technique to retrieve the vowel characters from our consonants-only or masked-vowels representations. This can be useful in NLP tasks where the output from an NLP task is English text to be used by humans.

The rest of the paper is organized as follows: in Section 2 we present related works. In Section 3 we present our text representations and the different tasks we conducted to compare the effectiveness of our approach. In Section 4, we present the experiments conducted, the results and the discussions on the results. In Section 5, we present the conclusions and discuss some possible future works. Finally, in Section 6, we present some of the limitations of our work.

## 2   Related Work

To the best of our knowledge, we did not find any work in the literature that introduces consonant-based text representations for NLP tasks. However, one of the prominent advantages of using these representations is the reduction in vocabulary size. Word-based embedding layer is usually a large layer due to the size of the vocabulary. For that reason, it has a high potential to suffer from overfitting. To address this issue, regularization techniques like embedding dropout can be applied (Gal and Ghahramani, 2016). On the other side, techniques such as dimensionality reduction (Raunak et al., 2019) quantization (Gholami et al., 2021) and distillation networks (Hinton et al., 2015) can be also be used. Although these methods are effective in addressing the issue of embedding overfitting, they do not address the issue by manipulating the vocabulary before embedding them. It would be an interesting to explore the model performance where these methods are combined with vocabulary reduction techniques.

Switching focus back to the issue of vocabulary size and reduction techniques, methods such as bag-of-words and TF-IDF were employed, traditionally. However, these approaches suffered from the issue of out-of-vocabulary (OOV) words and representing the words with huge sparse embeddings. While these methods excelled in modeling word relations, the OOV problem limited their use, particularly in natural language generation tasks. Additionally, they also faced the challenge of vocabulary explosion, where a large number of vocabulary items needed to be embedded, most of which had very few occurrences in the text (cf. Zipf's law (Zipf, 2016)). To overcome the limitations of sparse embeddings, dense-embedding representations, such as Word2Vec (Mikolov et al., 2013), were proposed that were more efficient and effective as compared to spare representations.

On the other hand, using characters as tokens do not suffer from the aforementioned issue. Moreover, tokenizing text as characters leads to a very small vocabulary size and no OOV issues. However, they do result in longer sequences, which can introduce bottlenecks such as vanishing and exploding gradients in sequence modeling models (Bengio et al., 1994). Character-based approaches are known to generate meaningless new words. Nonetheless, a promising approach is to first input characters into CNNs to extract features be-

fore feeding them into an RNN layer, as this has been shown to yield good results (Bradbury et al., 2017; Zhang et al., 2015; Conneau et al., 2017). This character-level approach has also been incorporated into transformer-based architectures (Ma et al., 2020; El Boukkouri et al., 2020).

Subword tokenization techniques were introduced as a trade-off between characters and words as tokens. These methods split a given word into chunks of characters called subwords. Numerous techniques have been proposed to achieve optimal subword splits for a given text. The main motivation behind this approach is to represent the text with a smaller vocabulary size of subwords without compromising performance on a given task. Some of these methods are language-specific, while others are data-driven (Alyafeai et al., 2022; Mielke et al., 2021). Data-driven methods are prevalent in recent language models such as the BERT family (Devlin et al., 2019), GPT family (Brown et al., 2020), and other large models. Examples of these methods include SentencePiece (Kudo and Richardson, 2018), WordPiece (Song et al., 2021), and UnigramLM (Kudo, 2018). Most of these methods are based on Byte-Pair-Encoding (BPE) data compression methods (Gage, 1994).

Another direction to mitigate the issues associated with tokenization is to utilize visual representation (Mansimov et al., 2020; Salesky et al., 2021). This approach has demonstrated robustness against noise and spelling mistakes. However, it is not widely adopted and may not be sufficiently competitive at its current stage of development (Mansimov et al., 2020).

## 3 Methodology

We introduce two variants of consonant-based representations for English text. The first representation involves removing all vowel letters (A, E, I, O, and U), regardless of their case, from the text, resulting in a consonants-only representation. This approach may exclude complete words that consist solely of vowels, such as the determiner 'A' and pronoun 'I'. However, such vowel-only words are rare in English. It is worth noting that certain NLP tasks require preserving the total number of words in the input sequence in the output sequence, as is the case with sequence labeling tasks such as POS tagging. For these tasks, this representation might be less suitable. This leads us to propose our second representation, where we mask all vowels with

Standing on the shoulders of giants

Stndng n th shldrs f gnts

St#nd#ng #n th# sh##ld#rs #f g##nts

Figure 1: An example of the proposed representations compared to the standard English text. The first line is the standard English text. The second line represents the consonants-only representation. The third line represents the masked-vowels representation.

the symbol '#'. Figure 1 illustrates an example of an English statement written in standard English as well as using our introduced representations.

### 3.1 Text analysis

We investigated the vocabulary reduction achieved by both of these representations compared to standard English text in a number of text corpora. In deep learning systems, the vocabulary size plays a crucial role in determining the embedding size and, in certain tasks involving text generation, the output layer size. This factor significantly impacts various aspects of the resulting model, including its overall size, the number of parameters involved, and training and inference times.

To assess the information loss resulting from the removal of vowels and evaluate the efficiency of the proposed representations, we examined their entropy (Shannon, 1951). Deep learning models learn their parameters by minimizing losses, such as cross-entropy, between their predictions and the expected output. If the entropy of these representations is similar to the entropy of standard English text, then, theoretically, they should be approximately as learnable as the standard English text. We computed entropy using the formula provided in Equation 1, where $t$ represents a token from the set of $T$ vocabulary in a given text.

$$H(t) = \sum_{t}^{T} -P(t) \log_2 P(t) \qquad (1)$$

### 3.2 Language modeling

To investigate language models, we trained three types of causal language models: statistical, RNN-based, and transformer-based models. The training corpus used for these models is the Wikitext-2 benchmark (Stephen et al., 2017).

For the statistical models, we trained (2 to 6)-gram models using the KenLM toolkit (Heafield, 2011). KenLM implements a modified Kneser-Ney smoothing technique (Heafield et al., 2013), which has been demonstrated to produce sequences with low perplexity. In all of our experiments, we maintained the default hyperparameters.

For the RNN-based language models, we constructed a network comprising an embedding layer, followed by a dropout layer (set to 0.333) (Gal and Ghahramani, 2016). This was then followed by 4 LSTM layers, each with 512 hidden units, and another dropout layer. The outputs were activated by a ReLU activation function and passed to a dense layer for prediction. We utilized cross-entropy with softmax activation as the loss function and employed the Adam optimizer for network optimization. We set the initial learning rate to 0.001 and decayed it by half when there was no improvement for an entire epoch. Training was performed for 100 epochs, and early stopping was implemented when there was no improvement in the validation loss for 5 consecutive epochs. Additionally, we employed output and embedding tying techniques similar to those described in (Press and Wolf, 2017; Inan et al.).

For the transformer-based language model, we implemented two decoder layers with two attention heads with dropout of 0.2. The feed-forward size is 200. The optimizer used is SGD with a decay factor of 0.25 if the validation loss does not improve for 1 epoch. The training stops if there is no improvement for 5 consecutive epochs. To overcome potential gradient explosion, we clipped gradient norms to 0.25.

For the RNN-based and the transformer-based models, we experimented with two tokenization schemes, word-based tokenization, as well as subwords tokenization using BPE implementation from sentencepiece (Kudo and Richardson, 2018).

### 3.3 NLP tasks

We expanded our investigation on these representations by comparing their performance with standard English text across various NLP tasks, employing different settings and configurations. These tasks include binary and multiclass text classification, POS tagging, NER and NMT. Language modeling serves as an upstream task for many downstream tasks such as spell correction, automatic speech recognition (ASR), machine trans-

lation, and optical character recognition (OCR). Therefore, it is crucial to evaluate the performance of the proposed representations on this task. Additionally, POS tagging and NER are examples of sequence labeling tasks, where the output has the same length as the input.

We emphasize that, in order to ensure a fair comparison, the proposed models were tuned to achieve the best results for standard text. Tuning the models to optimize the performance for the proposed representations might potentially yield superior results compared to standard text. However, this aspect was not explored in the scope of this study.

#### 3.3.1 Text classification

For text classification, we explored two types of tasks: binary sentiment analysis and multiclass text classification. For binary text classification, we utilized the IMDB reviews dataset (Maas et al., 2011). This dataset comprises 25,000 samples for training and 25,000 samples for testing. From the training set, we randomly allocated 10% of the samples for the validation set. For multiclass classification, we employed the AGNews benchmark (Zhang et al., 2015). This benchmark consists of four news classes, with 120,000 samples for training and 7,600 samples for testing. Similar to the IMDB dataset, we reserved 10% of the training set for validation.

We employed a similar architecture for both the datasets, consisting of a 4-layer bidirectional LSTM with 256 hidden units each and embedding layers of size 512. We applied a dropout rate of 0.4 to the embeddings layer and 0.333 to the LSTM layers. For the sentiment analysis task, we used binary cross entropy loss with sigmoid activation. For the multiclass text classification task, we utilized cross entropy loss with softmax activation. Similar steps were applied for cleaning and preprocessing in both experiments. In the cleaning phase, HTTP links, HTML tags, numbers, punctuation marks, emojis, and non-ASCII characters were removed from the text. The text was further processed by removing stopwords and lemmatizing the remaining words using the NLTK toolkit (Loper and Bird, 2002).

#### 3.3.2 Sequence labeling

Sequence labeling is a category of tasks where the input is a sequence of tokens, and the output is another sequence of tokens with the same length. These tasks, also known as token classification tasks, involve assigning a label to each token in

the input sequence, hence the name. In our study, we focused on two sequence labeling tasks: POS tagging and NER.

For POS tagging, we utilized the Universal Dependencies project (Nivre et al., 2020) tags. These tags consist of a total of 17 labels that can be used in multilingual settings. We trained our model on the English Web Treebank (EWT) dataset. The dataset comprises 254,825 words from 16,621 sentences. It has already been split into training, validation, and testing sets. For NER, we utilized the amended version of the CoNLL2003 dataset (Tjong Kim Sang and De Meulder, 2003), which was corrected by (Wang et al., 2019) and referred to as CoNLLpp. The CoNLLpp dataset consists of a total of 20,744 samples, with 14,041 samples allocated for training, 3,250 samples for validation, and 3,453 samples for testing. The dataset contains a total of 301,418 words. In this dataset, there are 9 different named entities. The model configuration used for training on the CoNLLpp dataset is similar to the one implemented for the text classification task, with the exception of the embedding dropout, which is set to 0.5. Additionally, the output size of the model is equal to the number of classes specific to each task.

### 3.3.3 Translation

Translation is an interesting task to study for this problem. We implemented a transformer-based translation model to translate from English (en) to German (de), and vice-versa. The tokenization method used for this experiment is word-based tokenization. To model the unknown tokens, we removed tokens that occur only once from the target sequence. The model is trained on Multi30K dataset (Elliott et al., 2016) which is an English-German image description parallel dataset. The dataset consists of 31.1K samples. 1K of these samples are reserved for testing, and another 1K samples are reserved for validation. We used the split provided by (Bentrevett). The employed architecture is a transformer model consisting of one encoder layer as well as one decoder layer with 8 multi-head attention, 2048 latent dimension, and 256 embedding size. The batch size used is 64. The optimizer used is RMSProb with sparse categorical cross-entropy loss. The model is trained for 10 epochs saving the checkpoint that achieved the lowest validation loss during training. The metric used to analyze the model performance is the 4-gram BLEU score.

### 3.4 Retrieving standard English representation from the proposed representations

We propose a sequence-labeling-based approach to retrieve the standard text from the proposed representations. This retrieval is crucial for tasks where human-readable text is required. We trained our system on the AGnews dataset, where the input texts are in our representations and the output text is in standard English. For the word-based sequence-labeling approach, we used models similar to those used in the text classification task. The models consist of 2 layers of bidirectional LSTMs with 512 hidden units each, and an embedding layer with a dropout of 0.25. The output layer is a softmax with the size equal to the vocabulary of the standard English text computed from the training set of the dataset.

For evaluating the performance of the models, we used the word error rate (WER) and character error rate (CER) as metrics. Additionally, to specifically analyze the performance on vowels, we introduced a new metric called the vowels error rate (VER). In this metric, we exclude the consonant characters while calculating the character error rate as mistakes can only be made on the vowel characters.

## 4 Results and Discussion

This section presents the results of the experiments conducted in this study. We begin by presenting the text analysis on the text corpora, followed by a discussion of the performance of the proposed representations compared to standard text in the context of language modeling. Next, we compare the performance of the proposed representations with standard English text across various NLP tasks. Additionally, we report the results on retrieval of the standard text representation from the proposed vowels-free representations. Finally, we conclude this section by providing further comparisons between these representations.

### 4.1 Text analysis

In Table 1, we present a summary of the vocabulary and token statistics at the word level for all the datasets (training sets) used in our experiments for the three text representations. The table reveals that dropping vowels results in a reduction of vocabulary size by at least 15% on the training sets, with an average reduction of 18%. This reduction

ratio tends to increase as the dataset size increases, as evidenced on the Wikitext dataset, where the reduction reaches approximately 23%. A similar pattern is observed for masked vowels, although the reduction is less significant, with a maximum of around 10% and an average of around 7%.

Regarding token size, the consonants-only representation experiences a decrease of approximately 4% caused by the omission of words consisting of vowels-only characters, except for the Multi30K dataset where this reduction is relatively large compared to other datasets. This is due to the nature of this dataset, as describing images extensively uses the "a" determiner. In fact, the determiner "a" and the pronoun "I" are the most prominent examples of such tokens. The smallest decrease in token size is observed in the Wikitext dataset, with only a 2% reduction. This discrepancy may be attributed to the fact that the dataset contains various characters besides English letters, including punctuation marks and characters from other languages. We intentionally did not preprocess this dataset by removing these characters to maintain its originality and reflect real-world language usage, as stated by the dataset authors (Stephen et al., 2017).

In Table 2, we present the entropy of the text at the word and character levels. In terms of word entropy, we observe that the consonants representation has the lowest entropy except for Multi30K due to the removal of the determiner "a" as previously discussed, followed by the masked-vowels representation, and then the standard English text. This result is expected considering the vocabulary size of each representation.

In terms of entropy at the character level, we observe a different pattern compared to the word level. The masked-vowels representation has the lowest entropy of all the three representations. This may seem counter-intuitive, but it can be explained by the fact that, on average, one or two consonant letters are followed by one or two vowels within a single word in the masked-vowels representation. By masking all vowels with a single character, the task of predicting the next character becomes easier, resulting in a reduction in the entropy of the entire text. The consonants-only representation exhibits a character entropy that is close to that of the standard English text, despite the fact that almost a fifth of the character set has been dropped (i.e., the 5 vowels out of the 26 characters). However, it should also be noted that the length of the text

is significantly reduced as vowels comprise more than a third of the text (approximately 38% in the datasets we are using).

## 4.2 Language Modeling

Table 3 presents the perplexity results for the language modeling (LM) tasks. From the table, we observe that, for n-gram LMs, the standard text has the highest perplexity, followed by the masked-vowels representation, and then the consonants-only representation. This pattern holds true across all the ngrams. The results are inline with the vocabulary sizes for the three representations. In contrast to the n-gram models, the RNN-based language models exhibit significantly lower perplexities, highlighting their power and capacity in language modeling tasks. Interestingly, there is a minimal difference in perplexity among the three representations in this experiment. In fact, the consonants-only representation shows slightly higher perplexity compared to the standard representation. This can be attributed to the absence of words like 'a' and 'I' in the consonants-only representation, which may pose a slight challenge in predicting the next words in their absence. The perplexity result for the standard text is the lowest for the transformer-based model showcasing the unique capabilities of transformers on this task. Following a similar pattern to RNN-based models, yet more notable, the difference between the three representations is almost negligible.

For the subwords as tokens, Table 4 presents the results for RNN-based and Transformer-based models. The vocabulary sizes in these experiments are 20,110, 13,041, and 17,247 for standard-text, consonants, and masked-vowels representations, respectively. The results of this experiment are inline with the results from the word-based tokenization. RNN-based and transformer-based models showcase even a smaller difference between the three text representations regardless of the notable difference in the vocabulary size.

## 4.3 NLP tasks

Table 5 provides an overview of the results obtained from sentiment analysis on the IMDB dataset and multiclass text classification on the AGNews dataset. The table indicates that the proposed representations exhibit competitive performance compared to the standard English text on both datasets. Notably, the consonants representation achieves slightly higher accuracy than the standard text in

| Text Corpora | Vocabulary size \|V\| | | | | | Tokens (N) | | |
|---|---|---|---|---|---|---|---|---|
| | $V_S$ | $V_C$ | $V_M$ | $V_C/V_S$ | $V_M/V_S$ | $N_S$ | $N_C$ | $N_C/N_S$ |
| Wikitext | 33,277 | 25,538 | 30,090 | 0.77 | 0.90 | 2,265,796 | 2,221,913 | 0.98 |
| IMDB | 280,617 | 231,543 | 260,846 | 0.83 | 0.93 | 5,844,680 | 5,613,725 | 0.96 |
| AGNews | 188,110 | 154,700 | 173,749 | 0.82 | 0.92 | 4,541,694 | 4,430,395 | 0.98 |
| EWT | 32,273 | 27,305 | 30,376 | 0.85 | 0.94 | 199,040 | 191,482 | 0.96 |
| CoNLLpp | 26,883 | 22,420 | 25,250 | 0.83 | 0.94 | 254,983 | 250,387 | 0.98 |
| Multi30k | 15,456 | 12,844 | 14,394 | 0.83 | 0.93 | 345,020 | 295,833 | 0.86 |

Table 1: Summary of the vocabulary and token statistics (at word level) comparing the proposed representations with the standard text. For Multi30K, we only considered the English portion of the dataset. (Please note: from here on, S represents the standard text, C represents consonants-only, and M represents masked-vowel representations)

| Corpora | $H_{wd}$ | | | $H_{ch}$ | | |
|---|---|---|---|---|---|---|
| | S | C | M | S | C | M |
| Wikitext | 10.2 | 9.8 | 10 | 4.8 | 4.7 | 4 |
| IMDB | 11.2 | 10.8 | 11 | 4.7 | 4.5 | 3.8 |
| AGNews | 12.1 | 11.6 | 11.8 | 4.9 | 4.7 | 4 |
| EWT | 11.1 | 10.6 | 10.8 | 4.8 | 4.6 | 3.9 |
| CoNLLpp | 10.8 | 10.4 | 10.7 | 5 | 4.9 | 4.2 |
| Multi30k | 8.9 | 9 | 8.5 | 4.4 | 4.1 | 3.5 |

Table 2: Text entropy at word and character levels on the training split of each dataset. $H_{wd}$ and $H_{ch}$ are the entropy at word and character levels, respectively. For Multi30K, we only considered the English portion of the dataset.

| Dataset | Accuracy | | |
|---|---|---|---|
| | S | C | M |
| IMDB | 85.17 | 83.68 | 84.68 |
| AGNews | 91.70 | 91.80 | 91.67 |

Table 5: A summary of the text classification results using the three representations.

| Task (Dataset) | Accuracy | | |
|---|---|---|---|
| | S | C | M |
| POS Tagging (EWT) | 91.44 | 87.26 | 91.16 |
| NER (CoNLLpp) | 93.97 | 93.88 | 94.54 |

Table 6: Summary of the results on the sequence labeling tasks.

| Model | Perplexity (PPL) | | |
|---|---|---|---|
| | S | C | M |
| 2-gram | 515.89 | 451.74 | 481.18 |
| 3-gram | 444.02 | 388.08 | 411.84 |
| 4-gram | 434.92 | 379.16 | 402.76 |
| 5-gram | 433.22 | 377.49 | 401.01 |
| 6-gram | 432.90 | 377.21 | 400.73 |
| RNN LM | 102.47 | 106.83 | 100.29 |
| Transformer LM | 94.90 | 94.60 | 95.79 |

Table 3: Perplexity results from the statistical n-grams and neural language models on the test set of Wikitext dataset.

| Model | Perplexity (PPL) | | |
|---|---|---|---|
| | S | C | M |
| RNN LM | 8.33 | 8.20 | 8.17 |
| Transformer LM | 7.52 | 7.50 | 7.52 |

Table 4: Language models perplexity results on sub-words tokenization

the AGNews dataset. These results suggest that vowels contribute minimal information in text classification tasks, and comparable performance can be achieved using the proposed representations, offering the advantages of smaller model size and faster training. Further discussions on this topic are presented in Section 4.5.

In Table 6, we present the results of the POS tagging and NER using the EWT and CoNLLpp datasets, respectively. The reported results are in terms of accuracy. In the POS tagging experiment, the consonants-only representation exhibits a significant decrease in performance by more than 4% compared to the standard text representation. However, the masked-vowels representation maintains a comparable performance to the standard text. To investigate the reasons behind the performance drop, a manual analysis was conducted using the confusion matrix. The analysis revealed that the consonants-only representation lacks words like 'a' and 'I', which constitutes a significant number of tokens. These words are easy to predict for POS tagging as determiner and pronoun, respectively.

| Task | BLEU | | |
|---|---|---|---|
| | S | C | M |
| English-to-German | 30.56 | 29.70 | 28.19 |
| German-to-English | 29.70 | 25.19 | 29.80 |

Table 7: Translation results in BLEU score

| | WER | CER | VER |
|---|---|---|---|
| Consonants | 8.99 | 4.06 | 5.60 |
| Masked-Vowels | 2.87 | 1.78 | 1.87 |

Table 8: Standard text retrieval results. WER is the word-error-rate, CER is the character-error-rate, and VER is the vowel-error-rate.

This constitutes the main reason for the drop in performance for the consonants-only representation.

NER, on the other hand, shows competitive performance compared to the standard English text. In fact, the masked-vowels representation outperforms the standard text representation by reducing the absolute error by over 0.5%, while consonants-only representation provides a comparable performance.

Table 7 presents the translation results from English to German (en-de) and vise-versa. Based on the results from the table, we can see that all the representations are giving us comparable results on the (en-de) translation task. For the (de-en) translation task, we can notice that the consonant-only approach witnesses a noticeable drop. This can be attributed mainly to the missing determinant "a" as discussed in Section 4.1. Interestingly, masked-vowels representation yields even better results than the standard English representation. It should be noted that the current model is optimally calibrated on the standard English representation. Tuning it on the proposed representations might lead to even better performances.

## 4.4 Retrieval of standard English representation from the proposed representations

In this section, we focus on the task of retrieving vowel information from the two vowel-free representations introduced in this study. We utilize the AGnews dataset for training and evaluating the models. Table 6 presents the results of standard text retrieval from the proposed representations, measured in terms of word-error-rate (WER), character-error-rate (CER), and vowels-error-rate (VER).

The table indicates that retrieving standard text from the masked-vowels representation yields superior results compared to the consonants-only representation. It is worth noting that part of the errors encountered during the retrieval from consonants-only representation can be attributed to the omission of words containing only vowels. However, we believe that such errors can be mitigated by im-

plementing a post-processing step, such as spelling and grammar correction, to refine the output.

To evaluate the model performance while disregarding words that consist solely of vowels, we calculate the aforementioned metrics after removing these tokens from both the predicted output and the original text. The resulting metrics are as follows: 6.4% for word-error-rate, 3.15% for character-error-rate, and 3.85% for vowels-error-rate. The approximately 3% difference in word-error-rate aligns with the reported ratio of consonant tokens to standard text tokens presented in Table 1 for the AGNews dataset.

To further investigate the errors in vowel retrieval from both the representations, we manually analyzed a subset of the first 100 samples from the test set. Our observations revealed the following insights:

- Masked-vowels representation: Some errors occurred due to the presence of unknown words—words in the test set that were not present in the training set. Since the model was not exposed to these words during training, it struggled to accurately retrieve the vowels. Additionally, we noticed that a significant portion of the errors stemmed from nouns. These nouns might have been relatively rare in the training set, resulting in lower model familiarity and increased difficulty in predicting the correct vowels for these words.

- Consonants-only representation: In the consonants-only representation, we observed additional sources of errors in vowel retrieval. These include errors relate to abbreviations and cases of letters, as well as challenges in capturing vowel positions within words.

## 4.5 Further comparisons

This section presents further comparisons of the proposed representations compared to the standard English text. Table 9 shows the model sizes of each dataset represented with our proposed representations compared to the standard English text.

| Dataset | Number of parameters (M) | | |
|---------|------|------|------|
|         | S    | C    | M    |
| Wikitext | 25.5 | 21.5 | 23.8 |
| IMDB     | 32.5 | 25.2 | 29.3 |
| AGNews   | 32.8 | 25.4 | 29.5 |
| EWT      | 16.4 | 14.3 | 15.6 |
| ConLLpp  | 18.4 | 16.4 | 17.7 |
| Multi30K | 16.2 | 15.0 | 15.7 |

Table 9: Models sizes for differed text representations in terms in the number of parameters (in Millions). For Multi30K, the reported results are for de-en experiment.

The table illustrates that the model size is closely related to the vocabulary size, as the embedding layers have a dominant effect on the model size. For example, the language models have sizes of 101 MB, 86 MB, and 95 MB respectively for the standard English, consonants-only, and masked-vowels representations. Additionally, the model size on storage devices is also influenced by its total number of parameters.

Moreover, a lower model size indicates a lower training time per epoch. In the language model experiment, for example, the training time per epoch is 82.14 seconds for the standard text, 67.55 seconds for the consonants-only representation, and 77.07 seconds for the masked-vowels representation. Thus, the reduced model size contributes to faster training.

Another aspect of comparison is the Out-Of-Vocabulary (OOV) rate and unknown tokens. In some tasks, it is common to remove words with a single occurrence to reduce model size and avoid overfitting. In the AGNews dataset, the number of single-occurrence vocabularies is 27,348 out of 79,037 for the standard text representation. For the consonants-only representation, the number of these vocabularies is 17,283 out of 54,596, and for the masked-vowels representation, it is 22,660 out of 67,982. Accordingly, the ratio of these unknown vocabularies are 34.60%, 31.66%, and 33.33% for the standard text, consonants-only representation, and masked-vowels representation, respectively.

## 5  Conclusion and future work

In this paper, we introduced two novel text representations based on English consonants. The first representation completely removes vowels from the input sequence, while the second representation masks vowels to a unique symbol. We conducted a comprehensive analysis of these representations by comparing their corpus statistics, including vocabulary and token sizes, text entropy, as well as the language model perplexity. Additionally, we evaluated the performance of these representations on various NLP tasks and compared the results with those obtained using the standard English text representation.

The experimental results demonstrated that the proposed representations achieved competitive performance without significant sacrifices. Notably, the models based on these representations exhibited smaller model sizes and a reduced number of parameters. This indicates the potential for more efficient training and deployment of models using these representations.

In future work, our focus will be on optimizing the architectures specifically for the proposed representations to achieve superior performances compared to the standard English text. We aim to evaluate the representations on a broader range of NLP tasks, such as text summarization and question answering, to assess their generalization abilities. Further investigation will be conducted to improve vowel retrieval.

## 6  Limitations

It is crucial to conduct additional experiments to evaluate the generalization capabilities of the proposed representations, particularly in important NLP tasks such as question answering and text summarization. Moreover, the present work would benefit from exploring larger and more diverse datasets to validate the effectiveness of the representations across different domains. Additionally, to improve the accuracy of retrieving the standard text from the consonants-only representation, it is recommended to investigate the use of an encoder-decoder architecture that can deal with missing vowels-only words, as the presented architecture was unable to deal with these missing words. This will contribute to enhancing the overall performance and applicability of the proposed representations.

## Acknowledgment

The authors would like to thank Saudi Data and AI Authority (SDAIA) and King Fahd University of Petroleum and Minerals (KFUPM) for supporting this work through SDAIA-KFUPM Joint Research Center for Artificial Intelligence grant number JRC–AI–RFP–06.

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
