# OpenReview forum: "Consonant is all you need: a compact representation of English text for efficient NLP"
_EMNLP/2023/Conference — EMNLP 2023 Findings_

### Official Review · Reviewer_bMmj · 2023-08-03

**Soundness:** 3

**Excitement:**

3: Ambivalent: It has merits (e.g., it reports state-of-the-art results, the idea is nice), but there are key weaknesses (e.g., it describes incremental work), and it can significantly benefit from another round of revision. However, I won't object to accepting it if my co-reviewers champion it.

**Paper Topic And Main Contributions:**

This paper proposes a method of representing English text with consonants for enabling to reduce vocabulary size, out-of-vocabulary rate, and training time. Specifically, they show two methods to remove vowels and mask vowels with a specific symbol.

The experiment results on several NLP tasks showed that the method using consonants gives comparable performances to using both consonants and vowels.

**Questions For The Authors:**

- A: It seems complicated to apply the proposed method to generate text for humans because the output tokens need vowels. Is this correct?

- B: Have the authors evaluated performances when using subwords?

- C: Is the same trend actual when using Transformer, which would be the current standard instead of LSTM?

**Reasons To Accept:**

- This paper shows a simple method that could be easily applied.
- The effectiveness has been validated on a variety of NLP tasks.

**Reasons To Reject:**

- The degree of effectiveness for reducing vocabulary size and number of model parameters is not so much.
- Since only one architecture was experimented with, the generality for other model architectures needs to be clarified.
- There are no evaluation results for using commonly used subwords in the neural language model. Subwords also are compact text representations that can reduce the vocabulary size and OOV rate like the proposed method.

**Reproducibility:**

4: Could mostly reproduce the results, but there may be some variation because of sample variance or minor variations in their interpretation of the protocol or method.

**Reviewer Confidence:**

3: Pretty sure, but there's a chance I missed something. Although I have a good feel for this area in general, I did not carefully check the paper's details, e.g., the math, experimental design, or novelty.

---

> ### Author Rebuttal · Authors · 2023-08-29
>
> We thank the reviewer for his/her efforts in reviewing the paper and for providing constructive comments and feedback. Below we provide a point-by-point rebuttal to the concerns raised:
>
> ***Comment #1:***
>
> The degree of effectiveness for reducing vocabulary size and number of model parameters is not so much.
>
> ***Reply to Comment #1***
>
> We thank the reviewer for the comment. According to table 1, the reduction of vocabulary size ranges from 15-23% for consonants-only representation (35% and 14% for consonant-only and masked-vowel subword representations, respectively). Furthermore, from Table 7, the model size reduction ranges from 11-22% for the consonants-only representation. These reductions might have significant implications in many tasks.
>
> ***Comment #2:***
>
> Since only one architecture was experimented with, the generality for other model architectures needs to be clarified.
>
> ***Reply to Comment #2:***
>
> We thank the reviewer for the comment. We have performed additional set of experiments using transformer models on the language modeling as well as machine translation tasks.
>
> We implemented a transformer-based language model on wikitext-2 dataset using both word and subword tokenization schemes. The updated results (PPL) are shown in the table below (part of Table 3 in the original manuscript):
>
> |     Tokenization    |     Model           |     Normal English    |     Consonants    |     Masked Vowels    |
> |---------------------|---------------------|-----------------------|-------------------|----------------------|
> |     Words           |     Transformers    |     94.9              |     94.59         |     95.79            |
> |                     |     LSTMs           |     105.61            |     108.44        |     103.56           |
> |     Subwords        |     Transformers    |     7.52              |     7.50          |     7.52             |
> |                     |     LSTMs           |     8.33              |     8.20          |     8.17             |
>
> Subword vocabulary size in:
> Standard English: 20,110,
> Consonants-only representation: 13,041, and
> Masked-vowels representation: 17,247.
>
> We can observe from that table that transformer-based language models show better perplexity as compared to other LMs. Moreover, both our text representations (with reduced character set) show similar perplexity as compared to the original representation thereby implying that little information is lost by using our modified representations.
>
> We performed machine translation tasks using Multi30k benchmark dataset [1]. Below are the results (4-gram BLUE scores) for two separate transformer-based translation models, one from German to English and the other one is from English to German to further explore the potential of the proposed representations.
>
> |  Translation Task | Normal English | Consonants | Masked Vowels |
> |:-----------------:|:--------------:|:----------:|:-------------:|
> | English-to-German |      30.56     |    29.70   |     28.19     |
> | German-to-English |      29.70     |    25.19   |     29.80     |
>
> Based on the results from the table, we can see that all the different representations are giving us comparable results on the English-to-German (en-de) translation task. For the German-to-English (de-en) translation task, we can notice that the consonants approach witnesses a noticeable drop. This might be the result of this representation lacking important words like the particle ‘a’ and the pronoun ‘I’. Interestingly, Masked-vowels representation on German-to-English (de-en) translation task yields even better results than the standard English representation. It should be noted that the current model is optimally calibrated on the standard English representation. Tuning it on the proposed representations might lead to a better performance.
>
> The details of experiments and the results will be added in the updated manuscript.
>
> ***Comment #3:***
>
> There are no evaluation results for using commonly used subwords in the neural language model. Subwords also are compact text representations that can reduce the vocabulary size and OOV rate like the proposed method.
>
> ***Reply to Comment #3***
>
> We added subwords tokenization for the language modeling task. As presented in a previous comment (Comment #3).
>
> ***Comment #4***
>
> A: It seems complicated to apply the proposed method to generate text for humans because the output tokens need vowels. Is this correct?
>
> ***Reply to Comment #4***
>
> Thanks for that comment. In fact, Section 4.4 of the manuscript presents a technique to retrieve vowels from our representation. Based on the conducted experiments with results shown in Table 6, we can notice that retrieving vowels gives promising results (CER ranging from 1.78-4.06). We will change the title of the section to better reflect the task.
>
> ***Comment #5***
>
> B: Have the authors evaluated performances when using subwords?
>
> ***Reply to Comment #5***
>
> We performed an additional set of experiments using subwords as tokenization as detailed in reply to Comment#2.
>
> ***Comment #6***
>
> C: Is the same trend actual when using Transformer, which would be the current standard instead of LSTM?
>
> ***Reply to Comment #6***
>
> We thank the reviewer for pointing this out. We performed additional sets of experiments using transformer models for language modeling and machine translation tasks as discussed above. Similar trends were observed.

---

### Official Review · Reviewer_6XDg · 2023-08-05

**Soundness:** 4

**Excitement:**

3: Ambivalent: It has merits (e.g., it reports state-of-the-art results, the idea is nice), but there are key weaknesses (e.g., it describes incremental work), and it can significantly benefit from another round of revision. However, I won't object to accepting it if my co-reviewers champion it.

**Paper Topic And Main Contributions:**

This paper presents English representation formats without vowels, and generated language models with the representations.  The experimental results show that the vowel-less representations have almost the same representation power and they work comparably with the model on the standard English in some application tasks.  The drop of vowels reduced the vocabulary size and it contributes to make a smaller models.

**Questions For The Authors:**

* A. What do you think about the words that have opposite meaning with and without vowel, such as asynchronous and asymmetry?
* B. Why the results of text retrieval task using the standard model are not shown in Table 6?

**Reasons To Accept:**

* A. An interesting findings in the vowel-less representation: in some situation it works comparably with the original English representation.
* B. Exhaustive experiments with multiple language modelings and multiple tasks.

**Reasons To Reject:**

* A. They tested the traditional n-gram models and neural models, but no results were shown on the recent methods of generative decoder models.
* B. The overall benefit of this method is not shown.  It is good if the trade-off between the accuracy and model size is compared with other distillation method like corpus shrinking and layer/dimension reduction.


**Reproducibility:**

3: Could reproduce the results with some difficulty. The settings of parameters are underspecified or subjectively determined; the training/evaluation data are not widely available.

**Reviewer Confidence:**

3: Pretty sure, but there's a chance I missed something. Although I have a good feel for this area in general, I did not carefully check the paper's details, e.g., the math, experimental design, or novelty.

**Typos Grammar Style And Presentation Improvements:**

A. LL532: Table 5 -> Table 6

---

> ### Author Rebuttal · Authors · 2023-08-29
>
> We thank the reviewer for his/her efforts in reviewing the paper and for providing constructive comments and feedback. Below we provide a point-by-point rebuttal to the concerns raised:
>
> ***Comment #1:***
>
> A. They tested the traditional n-gram models and neural models, but no results were shown on the recent methods of generative decoder models.
>
> ***Reply to Comment #1***
>
> We thank the reviewer for the comment. Based on the feedback, we implemented a transformer-based language model on wikitext-2 dataset using both word and subword tokenization schemes. The updated results are shown in the table below (part of Table 3 in the original manuscript):
>
> |     Tokenization    |     Model           |     Normal English    |     Consonants    |     Masked Vowels    |
> |---------------------|---------------------|-----------------------|-------------------|----------------------|
> |     Words           |     Transformers    |     94.9              |     94.59         |     95.79            |
> |                     |     LSTMs           |     105.61            |     108.44        |     103.56           |
> |     Subwords        |     Transformers    |     7.52              |     7.50          |     7.52             |
> |                     |     LSTMs           |     8.33              |     8.20          |     8.17             |
>
> Subword vocabulary size in:
> Standard English: 20,110,
> Consonants-only representation: 13,041, and
> Masked-vowels representation: 17,247.
>
> We can observe from that table that transformer-based language models show better perplexity as compared to other LMs. Moreover, both our text representations (with reduced character set) show similar perplexity as compared to the original representation thereby implying that little information is lost by using our modified representations. We plan to update our manuscript with these new experiments and results.
>
> ***Comment #2:***
>
> B. The overall benefit of this method is not shown. It is good if the trade-off between the accuracy and model size is compared with other distillation methods like corpus shrinking and layer/dimension reduction.
>
> ***Reply to Comment #2:***
>
> The overall benefits of the proposed representations come in many aspects like reductions in vocabulary size, OOV-rates, and training time. We will update our manuscript to further highlight these points.
>
> Regarding comparison with other distillation methods like corpus shrinking and layer/dimension reduction, we plan to add them in the related work section of the updated manuscript. For an empirical comparison, we will need to first reproduce the original results and then compare with our approach which was not possible in this short time. Moreover, it is worth noting that these techniques can also apply to our representations. It would be interesting to apply these methods on the proposed approach and investigate the results. We will update the manuscript to reflect this literature on the related work and future work.
>
> ***Comment #3:***
>
> A. What do you think about the words that have opposite meanings with and without vowels, such as asynchronous and asymmetry?
>
> ***Reply to Comment #3***
>
> This is an interesting observation. However, these aspects are not new to natural language processing. As an example, polysemous words like ‘mouse’ [computer device vs animal] and ‘bank’ [financial institute vs river bank vs use of bank as a verb] are common in English but are effectively captured from the context. Our results related to language modeling and the downstream tasks gives us an indication that this approach seems to work. Contextual embeddings similar to BERT shall be able to capture many of these situations.
>
> ***Comment #4***
>
> B. Why are the results of text retrieval tasks using the standard model not shown in Table 6?
>
> ***Reply to Comment #4***
>
> It seems there was some confusion related to the phrase “text retrieval”. By “text retrieval”, we meant retrieving the standard English representation from our proposed representations (vowel retrieval). We will rephrase the section title and captions in our updated manuscript to make it clearer.
>
> ***Comment related to Typos Grammar Style And Presentation Improvements***
>
> We thank the reviewer for pointing this out. We will correct the Table number in our updated manuscript.

---

### Official Review · Reviewer_yLpJ · 2023-08-09

**Soundness:** 2

**Excitement:**

3: Ambivalent: It has merits (e.g., it reports state-of-the-art results, the idea is nice), but there are key weaknesses (e.g., it describes incremental work), and it can significantly benefit from another round of revision. However, I won't object to accepting it if my co-reviewers champion it.

**Paper Topic And Main Contributions:**

This work presents an approach for reducing the size of embedding by representing text only with consonants, or by masking vowels.
They show that such representation would reduce the size of vocabulary and thus the size of embeddings more that the drop in prediction accuracy on text and token classification tasks.
The paper also presents a technique to retrieve vowel information from the processed representation to create human-readable text output in NLP applications.

**Questions For The Authors:**

see above pls

**Reasons To Accept:**

The proposed idea is interesting and straight forward to implement and can be easily integrated with existing NLP models and frameworks.
The paper is well-written and easy to read.

**Reasons To Reject:**

1) Reducing the vocabulary size is one way of reducing the size of embedding, however, there are other alternatives such as dimensionality reduction (Raunak et al. 2019), quantization (see some works here (Gholami et al. 2021)), bloom embedding (Serra & Karatzoglou 2017), distillation networks (Hinton et al. 2015), etc.
This work should be compared against some of these related baselines to show its true potential as an innovative approach for embedding compactness.

*Raunak, V., Gupta, V., & Metze, F. (2019, August). Effective dimensionality reduction for word embeddings. In Proceedings of the 4th Workshop on Representation Learning for NLP (RepL4NLP-2019) (pp. 235-243).

*Serrà, J., & Karatzoglou, A. (2017, August). Getting deep recommenders fit: Bloom embeddings for sparse binary input/output networks. In Proceedings of the Eleventh ACM Conference on Recommender Systems (pp. 279-287).

*Gholami, A., Kim, S., Dong, Z., Yao, Z., Mahoney, M. W., & Keutzer, K. (2021). A survey of quantization methods for efficient neural network inference. arXiv preprint arXiv:2103.13630.

*Hinton, G., Vinyals, O., & Dean, J. (2015). Distilling the knowledge in a neural network. arXiv preprint arXiv:1503.02531.

2) The perplexity experiments are carried out on obsolete language models (n-gram HMM, RNN) that are rarely used nowadays. To better align the paper with current NLP trends, I believe the authors should showcase their approach using transformer-based (masked) language models.

3) The reliance of this approach on a secondary step (vowel-retrieval) to make the text human-readable again could limit its applicability. It would be interesting to see how this representation would perform on generation tasks such as translation or summarization. Since the vowel-retrieval process is not loss-less (word-error-rate 9 for consonant-only and ~3 for masked-vowel representations), it may cause a drastic drop in the performance of the models on such tasks.

4) In addition, this extra vowel-retrieval step would add to the required computational steps and may actually increase the computational requirements (as opposed to the paper’s claim on saving on computational resources).

**Reproducibility:**

3: Could reproduce the results with some difficulty. The settings of parameters are underspecified or subjectively determined; the training/evaluation data are not widely available.

**Reviewer Confidence:**

3: Pretty sure, but there's a chance I missed something. Although I have a good feel for this area in general, I did not carefully check the paper's details, e.g., the math, experimental design, or novelty.

---

> ### Author Rebuttal · Authors · 2023-08-29
>
> We thank the reviewer for his/her efforts in reviewing the paper and for providing constructive comments and feedback.
> Below we provide a point-by-point rebuttal to the concerns raised:
>
> ***Comment #1:***
>
> Reducing the vocabulary size is one way of reducing the size of embedding, however, there are other alternatives such as dimensionality reduction (Raunak et al. 2019), quantization (see some works here (Gholami et al. 2021)), bloom embedding (Serra & Karatzoglou 2017), distillation networks (Hinton et al. 2015), etc. This work should be compared against some of these related baselines to show its true potential as an innovative approach for embedding compactness.
>
> *Raunak, V., Gupta, V., & Metze, F. (2019, August). Effective dimensionality reduction for word embeddings. In Proceedings of the 4th Workshop on Representation Learning for NLP (RepL4NLP-2019) (pp. 235-243).
> *Serrà, J., & Karatzoglou, A. (2017, August). Getting deep recommenders fit: Bloom embeddings for sparse binary input/output networks. In Proceedings of the Eleventh ACM Conference on Recommender Systems (pp. 279-287).
> *Gholami, A., Kim, S., Dong, Z., Yao, Z., Mahoney, M. W., & Keutzer, K. (2021). A survey of quantization methods for efficient neural network inference. arXiv preprint arXiv:2103.13630.
> *Hinton, G., Vinyals, O., & Dean, J. (2015). Distilling the knowledge in a neural network. arXiv preprint arXiv:1503.02531.
>
> ***Reply to Comment #1***
>
> We thank the reviewer for pointing us to important works done in this area. We believe that these techniques are complementary to what we are proposing, i.e., they can be applied after our text representation scheme which can potentially lead to further embedding size reduction. Furthermore, in order to apply this, we will first have to reproduce the original work which was not feasible in this short period of time. We will update our related work and other relevant sections to address this. As a future work, we should point to applying these techniques over our proposed text representations. Moreover, we would like to point out that reducing the embedding size is one of major goals but there are other aspects to our work such as reduction of OOV rate and training time.
>
> ***Comment #2:***
>
> The perplexity experiments are carried out on obsolete language models (n-gram HMM, RNN) that are rarely used nowadays. To better align the paper with current NLP trends, I believe the authors should showcase their approach using transformer-based (masked) language models.
>
> ***Reply to Comment #2:***
>
> We thank the reviewer for this valuable comment. We implemented a transformer-based language model on wikitext-2 dataset using both word and subword tokenization schemes. The updated results (PPL) are shown in the table below (part of Table 3 in the original manuscript):
>
> |     Tokenization    |     Model           |     Normal English    |     Consonants    |     Masked Vowels    |
> |---------------------|---------------------|-----------------------|-------------------|----------------------|
> |     Words           |     Transformers    |     94.9              |     94.59         |     95.79            |
> |                     |     LSTMs           |     105.61            |     108.44        |     103.56           |
> |     Subwords        |     Transformers    |     7.52              |     7.50          |     7.52             |
> |                     |     LSTMs           |     8.33              |     8.20          |     8.17             |
>
> Subword vocabulary size in:
> Standard English: 20,110,
> Consonants-only representation: 13,041, and
> Masked-vowels representation: 17,247.
>
> We can observe from that table that transformer-based language models show better perplexity as compared to other LMs. Moreover, both our text representations (with reduced character set) show similar perplexity as compared to the original representation thereby implying that little information is lost by using our modified representations. We plan to update our manuscript with these new experiments and results.
>
> ***Comment #3:***
>
> The reliance of this approach on a secondary step (vowel-retrieval) to make the text human-readable again could limit its applicability. It would be interesting to see how this representation would perform on generation tasks such as translation or summarization. Since the vowel-retrieval process is not loss-less (word-error-rate 9 for consonant-only and ~3 for masked-vowel representations), it may cause a drastic drop in the performance of the models on such tasks.
>
> ***Reply to Comment #3***
>
> As pointed out by the reviewer, vowel retrieval is a secondary step which needs to be applied in some applications but not all. Based on the reviewer’s feedback, we did perform machine translation tasks using Multi30k benchmark dataset [1]. Below are the results (4-gram BLUE scores) for two separate transformer-based translation models, one from German to English based on the reviewer feedback and the other one is from English to German to further explore the potential of the proposed representations.
>
> |  Translation Task | Normal English | Consonants | Masked Vowels |
> |:-----------------:|:--------------:|:----------:|:-------------:|
> | English-to-German |      30.56     |    29.70   |     28.19     |
> | German-to-English |      29.70     |    25.19   |     29.80     |
>
> Based on the results from the table, we can see that all the different representations are giving us comparable results on the English-to-German (en-de) translation task. For the German-to-English (de-en) translation task, we can notice that the consonants approach witnesses a noticeable drop. This might be the result of this representation lacking important words like the particle ‘a’ and the pronoun ‘I’. Interestingly, Masked-vowels representation on German-to-English (de-en) translation task yields even better results than the standard English representation. It should be noted that the current model is optimally calibrated on the standard English representation. Tuning it on the proposed representations might lead to a better performance. The details of experiments and the results will be added in the updated manuscript. Moreover, higher WER in our vowels-retrieval process can be reduced by using dictionary and spell-checking.
>
> [1]: Elliott, Desmond, et al. "Multi30K: Multilingual English-German Image Descriptions." Proceedings of the 5th Workshop on Vision and Language. 2016.
>
> ***Comment #4***
>
> In addition, this extra vowel-retrieval step would add to the required computational steps and may actually increase the computational requirements (as opposed to the paper’s claim on saving on computational resources).
>
> ***Reply to Comment #4***
>
> We thank the reviewer for pointing this out. Please note that this retrieval step should be applied to a selective set of tasks and not to every NLP tasks. Moreover, the retrieval system is a common system across all the tasks which should be trained only once.

---

### Meta-Review · Program_Chairs · 2023-09-24

**Recommendation:** 2

**Metareview:**

This paper proposes to represent English text with only consonants, in order to increase model efficiency, based on the assumption that consonants are more discriminative than vowels. The proposed idea is simple yet effective. However, Reviewers pointed out several problems in the experimental setup: 1) the paper currently lacks a comparison with alternative strategies to reduce the embedding matrix size, such as dimensionality reduction and quantisation (among others); 2) the neural architectures chosen for the language modelling experiment is obsolete and should include subword-based Transformers. (The Authors provided some first results in the rebuttal to this effect but these should be expanded to be significant). Moreover, the approach suffers from a major limitation, namely relying on retrieving back vowels to make generated text human-readable, which is error-prone and partly reduces efficiency gains.

---

### Decision · Program_Chairs · 2023-10-07

**Decision:**

Accept-Findings

**Comment:**

This paper proposes to represent English text with only consonants, in order to increase model efficiency, based on the assumption that consonants are more discriminative than vowels. The proposed idea is simple yet effective. However, Reviewers pointed out several problems in the experimental setup: 1) the paper currently lacks a comparison with alternative strategies to reduce the embedding matrix size, such as dimensionality reduction and quantisation (among others); 2) the neural architectures chosen for the language modelling experiment is obsolete and should include subword-based Transformers. (The Authors provided some first results in the rebuttal to this effect but these should be expanded to be significant). Moreover, the approach suffers from a major limitation, namely relying on retrieving back vowels to make generated text human-readable, which is error-prone and partly reduces efficiency gains.